# Retinal Ciliopathy in the Patient with Transplanted Kidney: Case Report

**DOI:** 10.3390/ijms23147582

**Published:** 2022-07-08

**Authors:** Ivona Bućan, Mirjana Bjeloš, Irena Marković, Diana Bućan

**Affiliations:** 1Eye Clinic, University Hospital Centre Split, 21000 Split, Croatia; irena.markovic@pmx.hr; 2Eye Clinic, Clinical Hospital Sveti Duh, 10000 Zagreb, Croatia; dr.mbjelos@gmail.com; 3Specialist’s Office of Occupational Medicine Diana Bućan, 21000 Split, Croatia; specordbucan@gmail.com

**Keywords:** RP1 gene, retinitis pigmentosa, retinal–renal ciliopathies

## Abstract

A review of a rare case of a proven mutation in the RP1 gene (RP1c.2029C>T, p. (ARG677*) in a kidney transplant patient was presented herein. According to his medical history, he had tonsillectomy performed at the age of 20 due to erythrocyturia, and at the age of 32 he was treated for malignant hypertension. The patient had been diagnosed with chronic renal failure at age 56 years. During an eye examination in 2016, retinitis pigmentosa was suspected and the patient was advised to run further tests. After an ophthalmological examination and tests, genetic testing was performed and a mutation in the RP1 gene encoding a family of proteins which are components of microtubules in photoreceptor primary cilia was proven. The literature search found that mutations in the RP1 gene have so far been exclusively associated with a non-syndromic form of retinal degeneration. However, the RP1 protein is expressed in the kidneys, and it remains unclear why the mutation of this gene so far was only specifically related to retinal photoreceptor function and not to arterial hypertension and renal disease. Primary cilia are thought to act as potential mechanosensory fluid-flow receptors in the vascular endothelium and kidney and their dysfunction results in atherosclerotic changes, hypertension, and chronic renal failure.

## 1. Introduction

Primary cilia are evolutionarily conserved, specialized microtubule-based cellular projections of nearly all human tissue and organs, with the exception of blood cells [1]. The primary cilium grows out of the mother centriole within the basal body which docks them to the cell. The tubules from basal body expand into the proximal segment of the primary cilium and form the microtubule backbone called axoneme, surrounded by the matrix and the membrane. The base of the primary cilium is separated from the proximal part by the transition zone which acts as the diffusion barrier and assembled intraflagellar transport system whose role is to control the movement of the variety protein complexes, which play important roles in maintaining the primary cilium [2]. The specifically localized receptors on the membrane of primary cilium sense extracellular signals and convert them into certain signaling pathways that control tissue and organ differentiation, development, and homeostasis [3,4]. As a result of the presence of primary cilia in retina and kidney, mutations in genes coding for primary ciliary proteins lead to the impairment of their structure or function, which may result in pleiotropic clinical manifestations in terms of various retinal and renal syndromes [2].

Part of the inherited retinal degeneration (IRD) includes retinal ciliopathies caused by structure disorder or dysfunction of specialized primary cilia that form the outer segment of photoreceptors [5]. Photoreceptor primary cilia are adapted for light detection by the presence of membranous discs above the transition zone containing visual pigments and other phototransduction proteins [6]. The structure of photoreceptor primary cilia is analogous to other primary cilia and a comprehensive proteomic study of mouse photoreceptor primary cilia identified around 2000 proteins, out of which hundreds were present in other primary cilia [7]. The most frequent cause of the IRDs is retinitis pigmentosa (RP). Retinitis pigmentosa is a heterogeneous group of hereditary disorders that result in progressive devolution of the retinal rods and cones leading to progressive loss of peripheral and central vision, affecting 1 in 3000–8000 people worldwide. RP has an autosomal dominant, autosomal recessive, and X-linked inheritance model and it can occur in a syndromic as well as non-syndromic nature [5,8,9].

Primary cilia have an important role in other organs in maintaining homeostasis. They act as mechanosensory organelles on the vascular endothelium and their dysfunction results in atherosclerotic changes and hypertension. In the nephron tubules and collecting ducts primary cilia regulate urine flow, osmolarity, and composition. Furthermore, defects in signaling pathways of primary cilia in kidney lead to renal impairment, which is the most common sign of primary ciliopathies [2]. The latter show pathology ranging from urinary concentration disorders in seemingly normal kidneys to those with cystic dysplasia. A kidney ultrasound can show normal-sized, small, and large kidneys due to cysts, with changes in echogenicity and abnormality in corticomedullary differentiation [10,11,12].

We present a case of retinitis pigmentosa in the patient with kidney transplant.

## 2. Case Report

A 63-year-old patient with a kidney transplant and a proven gene mutation that is pathological for retinitis pigmentosa was considered herein. The patient stated a positive family history of retinitis pigmentosa (his brother was affected with it as well), although he had not undergone genetic testing. According to his medical history, he had tonsillectomy performed at the age of 20 at the nephrologist’s recommendation due to erythrocyturia, and at the age of 32 he was treated for malignant hypertension (fundus examination showed disc edema on both eyes). The patient was diagnosed with chronic renal failure at age 56 years of age. A kidney ultrasound showed small hyperechogenic kidneys with the cysts and absent corticomedullary differentiation. Due to the further progression of renal disease, in 2016, renal replacement therapy was initiated by hemodialysis via a central venous catheter and then via peritoneal dialysis. In 2017, a cadaveric kidney transplantation was performed. He is currently under the supervision of a nephrologist for possible post-COVID-19 rejection of the transplanted organ. At the same time, when elevated values of renal parameters were found, problems with visual function also began. In the ophthalmological anamnesis, he stated that he had impaired vision in his both eyes, especially on the left. At the eye examination in 2016, retinitis pigmentosa was suspected, and the patient was advised to run further tests. Due to frequent dialysis treatments and preparations for kidney transplantation, he postpones further ophthalmic evaluation. However, after a certain period of time, further tests were carried out, such as the standard ophthalmological examination, spectral domain optical coherence tomography (SD-OCT) of the macula and optic disc (Zeiss, Nidek), visual field with central vision program (Octopus), electroretinogram, and, at the end of 2021, genetic testing. Genetic testing confirmed the diagnosis of retinitis pigmentosa; the patient is heterozygous for RP1c.2029C>T, p. (ARG677*) which is considered pathogenic, and heterozygous for GPR179 c.1368del, p. (Phe456Leufs*30), which is probably pathogenic.

All ophthalmological clinical findings and performed diagnostic tests with results are summarized in Table 1 and Table 2.

## 3. Discussion

The RP1 gene encodes a family of proteins that are considered to be structural and functional components in photoreceptor primary cilia that play an important role in the in the organization of the outer segment of rods and cones ensuring proper orientation and signaling [13,14]. The RP1 is a soluble protein that has been concentrated specifically in the newly formed disc membranes of the axoneme that harbor components of the visual signal transduction pathways at the junction of the outer segment and transition zone [14,15]. The true mechanism of disease development due to RP1 mutations is not completely understood and there are few possible causes. The RP1 mutation may result in abnormal connections between actin, cadherin, and myosin in the disc membranes, leading to their destabilization and death of photoreceptors. Additionally, the studies showed that possible mislocalization of rhodopsin in mice is due to mutated RP1 protein involved in transport of rhodopsin throughout the transition zone. These two possible causes of RP1 disease are exclusively related to photoreceptor function [16,17,18].

The RP1 gene located on chromosome 8 was first discovered in a family with autosomal dominant retinitis pigmentosa in southeastern Kentucky [13]. Individuals with heterozygous mutations in the RP1 gene can present with a preserved visual acuity until the late decades of life, while recessively inherited RP1 mutations may lead to severe form of the disease in early years of life [15]. The mentioned variant of the gene (RP1 c.2029C> T, p. (ARG677*) is a nonsense mutation responsible for adRP, but there are studies that confirm de novo mutation [19]. The most logical explanation for RP caused by truncated RP1 protein is a dominant negative mechanism in which the truncated RP1 protein competes for binding to axonemal microtubules and interferes with the stability of primary cilia. Location of adRP mutations in the last exon (4) of the RP1 gene supports the theory of truncated protein and dismisses the possible explanation of gain-of-function or novel toxic-effect [20]. The variation in the phenotype severity of RP1 disease in which the nonpenetrance or minimal disease expression in individuals with a proven heterozygous Arg677ter mutation suggests possible influence of unidentified genetic and/or environmental factors [15,21,22]. In the non-syndromic form of disease, de novo Arg677ter mutation supports the hypothesis of a mutational hotspot in the RP1 gene [19], while on the other hand studies have shown that de novo mutations are thought to play an important role in syndromic retinal ciliopathies involving intellectual disorders [23,24].

The RP1 gene shares limited homology with the neuron microtubule-associated protein-binding domains of doublecortin (DCX) and, as a result, the RP1 protein is the first photoreceptor microtubule-associated protein (MAP) to be identified [14]. So far RP1 mutations have been associated with isolated form of retinal degeneration without systematic diseases, but on the other hand the RP1 protein is expressed in the nephron distal tubules and collecting ducts [25,26]. The RP1 protein as a MAP that contains the DCX domains, which could control axoneme length and stability in vivo by promoting elongation or shortening of microtubules. The finding that a RP1 is a MAP provides the possible foundation for the pathophysiology of renal disease suggesting that the RP1 protein may have role in regulating the axoneme length and stability of primary cilia in kidney [14,16,17].

In the last decade, there have been a multitude of clinical studies for IRDs and in the end of 2017, the Luxturna, the first genetic therapy for a patients with biallelic variations of RPE65 was approved. Therefore, when retinitis pigmentosa is suspected, the key is to undergo genetic testing to determine underlying mutation. The therapy is based on subretinal injection of adeno-associated vector delivering a healthy copy of RPE65 gene and enabling viable retinal cells to produce a missing enzyme. Effects of the treatment are estimated by visual acuity, visual field, light sensitivity, and the Multi-Luminance Mobility Test in which patients move around obstacles under different light levels to measure functional vision in daily life activities [27,28]. Many new treatments are still being tested such as optogenetic therapy for late stage of RP. This therapy combines a subretinal injection of gene encoding the optogenetic sensor which is a light-gated cation channel that can be stimulate with the light via high-tech goggles. Optogenetic therapy using a light to activate retinal cells could be helpful for all types of RP [28]. Furthermore, the most promising treatment for IRDs is intravitreal transplantation of stem cells [29]. The mutations in the RP1 gene are responsible for 3–5% of autosomal dominant retinitis pigmentosa cases. To this day, we do not yet have adequate therapy for diseases caused by RP1 mutations. Cases in which heterozygous family members do not suffer from RP symptoms show that one healthy allele of the RP1 gene can prevent the retinal degeneration of rods and cones. Therefore, the possible future of treating retinal degeneration caused by a truncated RP1 protein is a gene therapy that silences the expression of the mutated allele [30].

## 4. Conclusions

Our report illustrated a rare case of a proven mutation in RP1 gene (RP1c.2029C> T, p. (ARG677*), which has so far been exclusively associated with an isolated form of retinal degeneration, in a patient who has systematic diseases, hypertension, and chronic renal failure. Understanding these pathological conditions can help ophthalmologists or other doctors recognize the connection between seemingly unrelated diseases and suspect the presence of systemic disease. Additionally, understanding the pathophysiology of disease in kidney and the eye may lead to the development of new clinical examination protocols and treatment strategies for both renal and ocular disease.

## Figures and Tables

**Figure 1 ijms-23-07582-f001:**
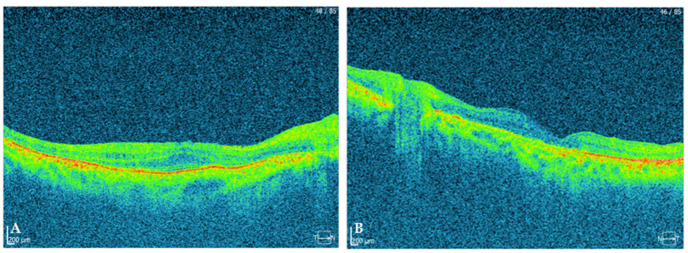
(**A**) SD-OCT image of the macula of the right eye; (**B**) SD-OCT image of the macula of the left eye.

**Figure 2 ijms-23-07582-f002:**
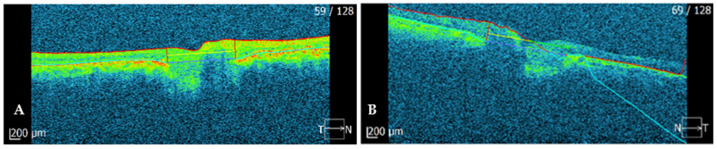
(**A**) SD-OCT image of the optic disc of the right eye; (**B**) SD-OCT image of the optic disc of the left eye.

**Figure 3 ijms-23-07582-f003:**
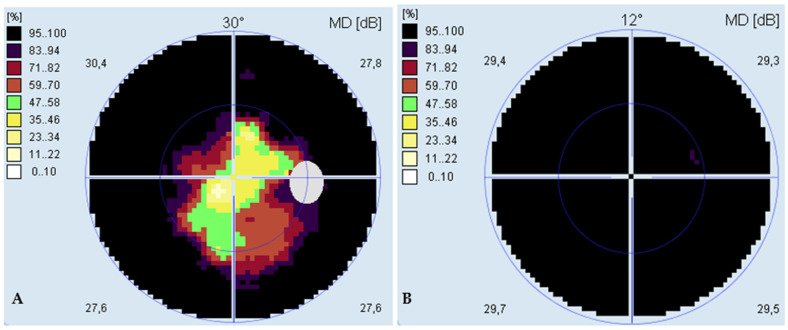
(**A**) Visual field of the right eye; (**B**) visual field of the left eye.

**Figure 4 ijms-23-07582-f004:**
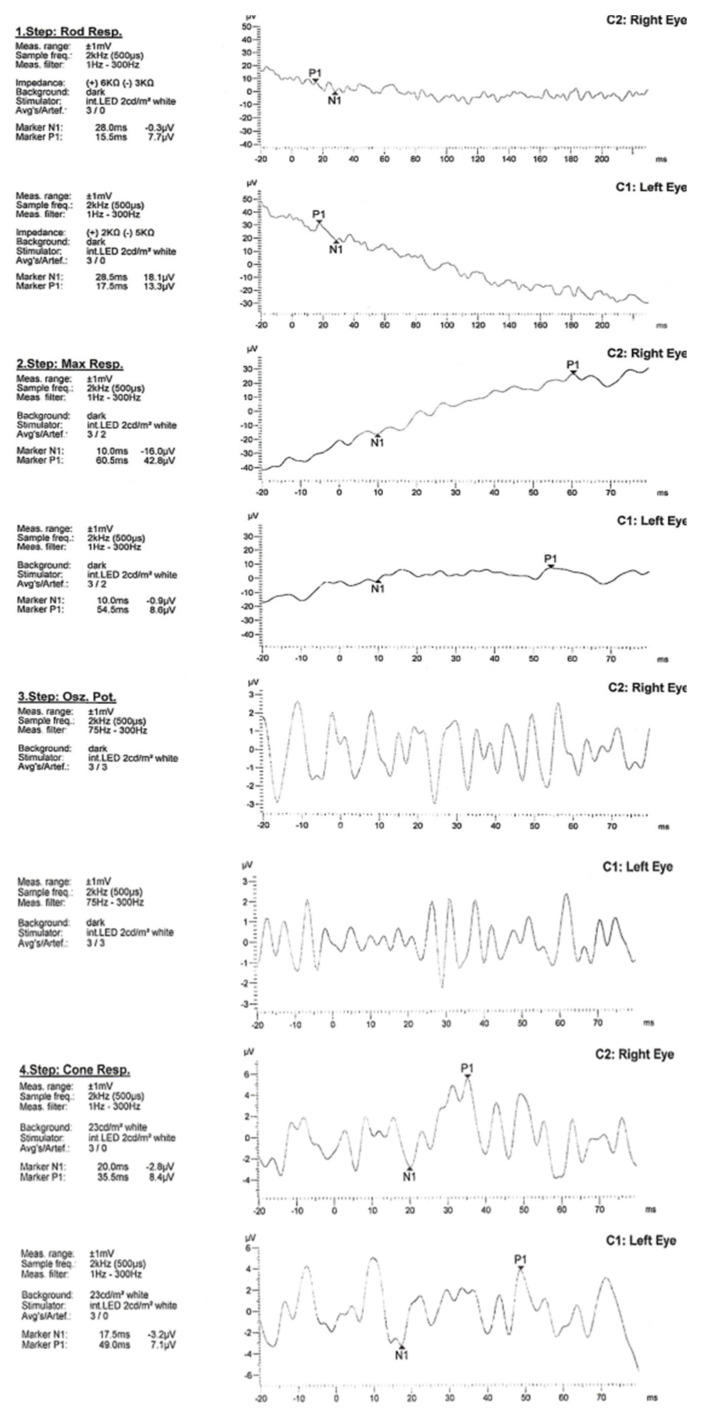
Electroretinographic examination.

**Table 1 ijms-23-07582-t001:** Ophthalmological clinical findings.

	Right Eye	Left Eye
**Best corrected visual acuity**	0.4	light perception with projection
**Intraocular pressure**	15 mmHg	15 mmHg
**Anterior segment**	posterior subcapsular cataract; reduced pupil response	macular corneal opacity; posterior subcapsular cataract; reduced pupil response
**Fundus**	waxy disc pallor; attenuation of retinal vessels; bone spicule-shaped pigment deposits; structural changes in fovea	wavy disc pallor; attenuation of retinal vessels; bone spicule-shaped pigment deposits; structural changes in fovea

**Table 2 ijms-23-07582-t002:** Ophthalmological diagnostic tests.

	Right Eye	Left Eye
**SD-OCT (macula)**	disrupted and thinned inner/outer segment of photoreceptors and retinal pigment epithelium (Figure 1A)	disrupted and thinned inner/outer segment of photoreceptors and retinal pigment epithelium (Figure 1B)
**SD-OCT (optic disc)**	flattened optic nerve head with thinned optic nerve (Figure 2A)	flattened optic nerve head with thinned optic nerve (Figure 2B)
**Computerized perimetry**	preserved central 10⁰ visual field (Figure 3A)	absolute scotoma (Figure 3B)
**Electroretinogram**	delayed implicit time and reduction in amplitude of the scotopic rod, photopic con and flicker response (Figure 4)	severely delayed implicit time and reduction in amplitude of the scotopic rod, photopic con and flicker response (Figure 4)

## Data Availability

The data presented in this case report are available on request from the corresponding author. The data are not publicity available due to privacy protection.

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
