# Peer review of "Retinal Ciliopathy in the Patient with Transplanted Kidney: Case Report"

_ijms, 2022, doi:10.3390/ijms23147582_

Round 1
Reviewer 1 Report
The manuscript titled ‘‘Retinal ciliopathy in the patient with transplanted kidney: case report’’ presents a case of retinitis pigmentosa in a patient with a kidney transplant. The authors describe a case report of a patient with chronic retinal failure and proved gene mutations associated with retinitis pigmentosa.
Concerns
1. The authors write both “cilia” and “primary cilia” throughout the document. Fort the sake of clarity, the authors should use a single term, i.e. cilia can be either motile or non-motile. The Primary cilium is non-motile.
2. The message of the article is somewhat vague. Please clarify.
3. The figures could benefit from a better organization.
Author Response
Re: Response to review
Manuscript ID: ijms-1775258
Retinal ciliopathy in patient with transplanted kidney: case report
Dear,
We would like to thank the assistant editor and reviewers for careful and thorough reading of this manuscript and for the thoughtful comments and constructive suggestions, which help to improve the quality of this manuscript. The corresponding changes and refinements made in the revised manuscript are summarized in our point-by-point responses below.
Reviewer #1:
The manuscript titled ‘‘Retinal ciliopathy in the patient with transplanted kidney: case report’’ presents a case of retinitis pigmentosa in a patient with a kidney transplant. The authors describe a case report of a patient with chronic retinal failure and proved gene mutations associated with retinitis pigmentosa.
Concerns
1. The authors write both “cilia” and “primary cilia” throughout the document. Fort the sake of clarity, the authors should use a single term, i.e. cilia can be either motile or non-motile. The Primary cilium is non-motile.
Response: We changed "cilia" to "primary cilia" in the entire manuscript.
2. The message of the article is somewhat vague. Please clarify.
Response: We re-writed the conslusion to be more specific about the reason behind our decision to illustrate this case report and the importance of case in clinical practise.
3. The figures could benefit from a better organization.
Response: In the section titled Case Report we created the two tables – the first one summarises ophthalmological clinical finding, whereas the second table outlines diagnostic tests and their results. These two tables were created as a suggestion of other reviewer. We provide a better quality and better organization of figures (the figures are placed side by side). As for the images labeled Fig. 2a and Fig. 2b instead of the numerical values of the OCT imaging of the optic disc, we changed the figure to an OCT representation in which the altered optic disc are clearly visible. For the Fig.3a and Fig,3b, instead of all additional numerical values of visual field, we only use the grayscale display of visual field which best presents the changes and loss of the visual field.
We reprashed the long sentences (marked in blue) that had similarity with other papers. We submited an informed consent and additional approval of the owner and main ophthalmologist of private Eye Polyclinic. The manuscript was checked by M.Ed. in English language and literature.
We will be happy to make further adjustments, if necessary.
Sincerely,
Ivona Bućan and co-authors
Reviewer 2 Report
This manuscript is a case report on a patient with mutation in RP1 gene and who has retinal ciliopathy and renal failure.
Comments.
1. Introduction page 1, line 9. Cilia functions are explained and exemplified by "trafficking", Is that protein-trafficking? The term trafficking is not mainstream in cell biology.
3. Introduction page 1, line 12. It is stated that primary cilia are in nearly all tissues and organs.. This statement should be modulated and more specific even though this statement then leads to that the functions of cilia are pleiotropic.
4. Introduction page 1, lane 19. od should read of.
5. Case report. A table that summarises the clinical findings and investigation methods would facilitate for the reader in section 2.
6. It would be interesting to read a paragraph on what treatments were offered and used with intended effects. It would also be interesting to read about what additional potential treatments have been considered and the discussion on the rationale for their use.
7 Discussion page 8, para 4, line 3. Statement that retina and kidney share developmental pathways. This should be rephrased. There is very little or anything that is shared between the development of retina and kidney and the comparison seem to be constructed. The discussion ends up in that both structures have cells that are dependent on primary cilia. This is correct but not a specific feature and the authors have concluded in the intro that there are cilia in nearly all tissues and organs.
The logics in this part of the discussion is therefore limping. It is probably true that the reason for that this RP1 ciliopathy manifests in retina and in kidney is that both organs have cells with a primary cilium. The discussion should increase specificity and highlight why this case is described and interesting.
8. Conclusion. The conclusion is very vague and is not addressing the specific clinical case in this case report. The conclusion rather gives general statements that many ciliopathies have overlapping phenotypes and that studies of this disease would contribute to a better understanding of genotype-phenotype of ciliopathies.
It is suggested that the conclusion should be based on specific data derived from the presented case and what insights this particular case gives on to the understanding of genotype-phenotype. It would be valuable to give an opinion how general or specific the phenotype of the described case is in relation to other similar cases.
Author Response
Re: Response to review
Manuscript ID: ijms-1775258
Retinal ciliopathy in patient with transplanted kidney: case report
Dear,
We would like to thank the assistant editor and reviewers for careful and thorough reading of this manuscript and for the thoughtful comments and constructive suggestions, which help to improve the quality of this manuscript. The corresponding changes and refinements made in the revised manuscript are summarized in our point-by-point responses below.
Reviewer #2:
This manuscript is a case report on a patient with mutation in RP1 gene and who has retinal ciliopathy and renal failure.
Comments.
1. Introduction page 1, line 9. Cilia functions are explained and exemplified by "trafficking", Is that protein-trafficking? The term trafficking is not mainstream in cell biology.
Response: In Introduction, line 9 is referred to the fact that protein-trafficking is important for the function of cilia. This sentence has been reformulated that the intraflagellar system controls the movement of protein complexes which is essential for ciliary function.
3. Introduction page 1, line 12. It is stated that primary cilia are in nearly all tissues and organs.. This statement should be modulated and more specific even though this statement then leads to that the functions of cilia are pleiotropic.
Response: Due to the fact that, in the Introduction section, line 2 and line 12 were two similar sentences, the sentence in line 2 has been reformulated. Furthermore, in the introduction from line 13 to 15, it is specified that because of the presence of primary cilia in the retina and kidney, mutations in genes encoding ciliary proteins result in retinal and renal diseases.
4. Introduction page 1, lane 19. od should read of.
Response: This is now corrected, as suggested.
5. Case report. A table that summarises the clinical findings and investigation methods would facilitate for the reader in section 2.
Response: In the section titled Case Report we created the two tables – the first one summarises ophthalmological clinical finding, whereas the second table outlines diagnostic tests and their results. We provide a better quality and better organization of figures (the figures are placed side by side) as a suggestion of other reviewer. As for the images labeled Fig. 2a and Fig. 2b instead of the numerical values of the OCT imaging of the optic disc, we changed the figure to an OCT representation in which the altered optic disc are clearly visible. For the Fig.3a and Fig,3b, instead of all additional numerical values of visual field, we only use the grayscale display of visual field which best presents the changes and loss of the visual field.
6. It would be interesting to read a paragraph on what treatments were offered and used with intended effects. It would also be interesting to read about what additional potential treatments have been considered and the discussion on the rationale for their use.
Response: In section Discussion we added a paragraph about existing and potential treatments.
7. Discussion page 8, para 4, line 3. Statement that retina and kidney share developmental pathways. This should be rephrased. There is very little or anything that is shared between the development of retina and kidney and the comparison seem to be constructed. The discussion ends up in that both structures have cells that are dependent on primary cilia. This is correct but not a specific feature and the authors have concluded in the intro that there are cilia in nearly all tissues and organs. The logics in this part of the discussion is therefore limping. It is probably true that the reason for that this RP1 ciliopathy manifests in retina and in kidney is that both organs have cells with a primary cilium. The discussion should increase specificity and highlight why this case is described and interesting.
Response: We changed the paragraph 4 in Discussion section to be more specific. We stated that since RP1 protein has MAP function (compared to the other two functions of RP1 proteins that reflect the stability of photoreceptor membrane discs and rhodopsin transport related to photoreceptor function), which means that it has a role in regulating primary cilia length and stability, the disorder of its function may be the foundation of the pathophysiology of renal disease. It is also explained why this case was presented given that RP1 mutations have so far been associated with isolated retinal disease without systemic diseases.
8. The conclusion is very vague and is not addressing the specific clinical case in this case report. The conclusion rather gives general statements that many ciliopathies have overlapping phenotypes and that studies of this disease would contribute to a better understanding of genotype-phenotype of ciliopathies. It is suggested that the conclusion should be based on specific data derived from the presented case and what insights this particular case gives on to the understanding of genotype-phenotype. It would be valuable to give an opinion how general or specific the phenotype of the described case is in relation to other similar cases.
Response: We re-writed the conslusion to be more specific about the reason behind our decision to illustrate this case report and the importance of case in clinical practise.
We reprashed the long sentences (marked in blue) that had similarity with other papers. We submited an informed consent and additional approval of the owner and main ophthalmologist of private Eye Polyclinic. The manuscript was checked by M.Ed. in English language and literature.
We will be happy to make further adjustments, if necessary.
Sincerely,
Ivona Bućan and co-authors
Round 2
Reviewer 1 Report
The authors answered all questions with valid points. The authors have made an effort to improve their manuscript and the current form reads better than the previous version.